Journal of Data-centric Machine Learning Research (2025)          Submitted 01/25; Revised 06/25; Published 06/25

# Chronicling Germany: An Annotated Historical Newspaper Dataset

**Christian Schultze**
*High-Performance Computing and Analytics (HPCA-Lab) Universität Bonn*          S6CNSCUL@UNI-BONN.DE

**Niklas Kerkfeld**
*HPCA-Lab, Universität Bonn*          S6NIKERK@UNI-BONN.DE

**Kara Kuebart**
*Institut für Geschichtswissenschaft Universität Bonn*          KARA.KUEBART@UNI-BONN.DE

**Princilia Weber**
*Institut für Geschichtswissenschaft, Universität Bonn*          S5PRWEBE@UNI-BONN.DE

**Moritz Wolter**[*]
*HPCA-Lab, Universität Bonn*          MORITZ.WOLTER@UNI-BONN.DE

**Felix Selgert**[*]
*Institut für Geschichtswissenschaft, Universität Bonn*          FSELGERT@UNI-BONN.DE

**Reviewed on OpenReview:** HTTPS://OPENREVIEW.NET/FORUM?ID=YHDFQVTYE9

**Editor:** Hugo Jair Escalante

## Abstract

The correct detection of dense article layout and the recognition of characters in historical newspaper pages remains a challenging requirement for Natural Language Processing (NLP) and machine learning applications in the field of digital history. Digital newspaper portals for historic Germany typically provide Optical Character Recognition (OCR) text, albeit of varying quality. Unfortunately, layout information is often missing, limiting this rich source's scope. Our dataset is designed to enable the training of layout and OCR models for historic German-language newspapers. The Chronicling Germany dataset contains 801 annotated historical newspaper pages from the time period between 1617 and 1933. The paper presents a processing pipeline and establishes baseline results on in- and out-of-domain test data using this pipeline. Both our dataset and the corresponding baseline code are freely available online. This work creates a starting point for future research in the field of digital history and historic German language newspaper processing. Furthermore, it provides the opportunity to study a low-resource task in computer vision.

**Keywords:** Digital History, Machine Learning, Optical Character Recognition (OCR)

## 1 Introduction

Newspapers are essential sources of information, not just for modern readers, but particularly in the past when other communication channels like the internet or radio were not yet available. Even more importantly, historical newspapers allow historians and social scientists to study social groups' opinions and cultural values and to use information from newspapers in causal models (for more details, see A.2). This paper presents the *Chronicling Germany* -dataset, consisting of 801 annotated high-resolution scanned newspaper pages from the period between 1617 and 1933.

---

[*]. Equal supervision

With the emergence of digital newspaper portals, using historical newspapers has become easier in recent years [1]. These portals provide text via OCR but often lack reliable layout information for their German language content, which is essential for digital history applications, many of which would require newspaper articles to be treated as individual documents. Our dataset will help to reduce the character error rate and aims at considerably improving the detection of individual elements of a newspaper page, like articles or single advertisements. The former is important to prevent algorithms from connecting unrelated text regions and preserve the order in which text regions should be read. To this end, the text layout is systematically annotated using nine classes.

From a computer science view, a collection of successful approaches allows us to process modern documents (Blecher et al., 2023; Davis et al., 2022). For historical documents, large-scale data sets exist (Dell et al., 2024) but are mostly focused on English language material set in Antiqua-like typefaces. For continental European languages, existing datasets are much smaller (Abadie et al., 2022; Nikolaidou et al., 2022; Kodym and Hradis, 2021; Clausner et al., 2015)

Until more annotated data becomes available, the processing of historical continental European newspaper pages is, therefore, a low-resource task, highlighting the need for more data. While low-resource tasks are well-established in natural language processing (Adams et al., 2017; Fadaee et al., 2017; Hedderich et al., 2021; Zoph et al., 2016) , low-resource settings remain under-explored in computer vision (Zhang et al., 2024) . Historical German newspapers are interesting in this context due to their dense layout (see also Supplementary Figure 6 ) and the Fraktur font. Fraktur differs significantly from the Antiqua typefaces that dominate modern Western texts. To the contemporary eye, Fraktur letters appear dense, which also impacts layout recognition. Furthermore, in addition to the font, our dataset features the archaic 'long s' or ' ſ ', which is no longer used today. The 'sz' or 'ß' is specific to the German language and also appears in the data. Historically, it emerged when the common combination 'ſz' merged into a single letter 'ß', unlike the 'long s' it still appears in contemporary texts. The aforementioned differences limit our ability to transfer existing solutions designed for modern documents or English-language historical newspapers. This motivates the collection of additional data.

The task of processing German newspapers is also highly relevant to historians. Especially in the 19th century, local communities, interest groups, and political parties created their own newspapers. The *Deutsche Zeitungsportal*[2] counts 698 German newspapers in 1780, this number rose to over 14,000 in 1860 and peaked at 50,848 papers in 1916 (see Figure 1).

Plenty of digitized pages allow researchers to systematically search for cultural values and historical change. Unfortunately, most of the pages available on such platforms contain either no or incorrect layout information. With text lines being in disarray, the pre-processed text data from these pages cannot be analysed computationally. Additionally, untrained modern human readers struggle with font differences, limiting the usefulness of unprocessed data to researchers lacking this specific skill. Thus, creating a pipeline capable of accurately processing this vast amount of data to a format readable to both a machine and a researcher without specific language and typeface skills is an important step in making these resources accessible. Furthermore, the availability of machine-readable newspaper archives is valuable to social scientists who recently started to use historical newspapers to track treatment variables or measure the impact of institutions or policies on social life (Beach and Hanlon, 2023).

Additionally, the layout of German historical newspapers is often complex, consisting of several columns, multiple horizontal sections and up to 500 elements to annotate per page. To create this dataset, eleven student assistants with a background in history have spent a total of 2,700 hours annotating the layout of 801 pages. These include approx. 1,900 individually annotated advertisements that consist of approx. 5,700 polygon regions. We also provide ground truth text annotations, which are not as costly since we start from network-generated OCR -output and correct

---

1. For Germany, e.g., the *Deutsche Zeitungsportal* ( `https://www.deutsche-digitale-bibliothek.de/newspaper/`) and *zeit.punkt NRW* (`https://zeitpunkt.nrw/`)
2. https://www.deutsche-digitale-bibliothek.de/newspaper/

Table 1: Datasheet listing newspaper names, page counts, lines, words and polygon-regions. As layouts change in a paper's history, we include issues from different years. In such cases a paper appears more than once in the table. Newspapers that have been used as out-of-distribution test data only are marked with *

| Year | Newspaper | Pages | Lines | Words | Regions |
|---|---|---|---|---|---|
| 1617 | Newspaper with no title, 30-Years war | 8 | 268 | 2,535 | 24 |
| 1626 | Newspaper with no title, 30-Years war | 8 | 276 | 2,618 | 28 |
| 1700 | Reichs Post Reuter | 8 | 350 | 2,991 | 42 |
| 1748 | Leipziger Zeitung | 4 | 292 | 2,051 | 56 |
| 1785 | Holzmindensches Wochenblatt | 8 | 238 | 1,722 | 31 |
| 1785 | Schwäbischer Merkur | 11 | 1017 | 7,060 | 77 |
| 1813 | * Donau Zeitung | 4 | 304 | 1,706 | 47 |
| 1813 | Königlich privilegierte Stuttgarter Zeitung | 4 | 361 | 2,271 | 46 |
| 1834 | Fränkischer Kurier | 4 | 412 | 3,130 | 62 |
| 1849 | * Hamburgischer unpartheiischer Correspondent | 4 | 2,250 | 15,711 | 167 |
| 1851 | Ostpreussische Zeitung | 4 | 1,183 | 9,538 | 126 |
| 1852-1888 | Special Issues Kölnische Zeitung | 20 | 826 | 8,063 | 134 |
| 1856 | Der Bazar | 11 | 1,333 | 10,503 | 114 |
| 1857 | Berliner Börsen Zeitung | 5 | 1,189 | 8,048 | 169 |
| 1866 | Bonner Zeitung | 4 | 1,400 | 10,364 | 176 |
| 1866 | Fränkischer Kurier | 6 | 1,814 | 12,375 | 302 |
| 1866 | Kölnische Zeitung | 420 | 248,295 | 2,137,972 | 17,050 |
| 1866 | Neue Berliner Musikzeitung | 8 | 1,036 | 8,261 | 116 |
| 1866 | Pfälzer Zeitung | 4 | 1,080 | 7,666 | 127 |
| 1866 | Vossische Zeitung | 31 | 4,671 | 34,057 | 1,129 |
| 1866 | Weisseritz Zeitung | 8 | 832 | 5,795 | 114 |
| 1867 | Hannoverscher Courier | 4 | 1,946 | 13,905 | 226 |
| 1867 | Neue preussische Zeitung | 4 | 3,179 | 23,067 | 203 |
| 1870 | Weisseritz Zeitung | 8 | 767 | 4,950 | 187 |
| 1871 | * Karlsruher Zeitung | 4 | 1,308 | 9,440 | 131 |
| 1889 | Mode und Haus, Illustrirte Kinderwelt | 8 | 342 | 2,976 | 66 |
| 1891 | Bonner Zeitung | 4 | 1,469 | 9,690 | 406 |
| 1898 | Dresdner Journal | 8 | 3,744 | 26,657 | 370 |
| 1917 | Muenchner Neue Nachrichten | 6 | 3,421 | 19,292 | 621 |
| 1918 | * Darmstädter Zeitung | 4 | 1,388 | 9,309 | 207 |
| 1923 | Eibenstocker Tagblatt | 4 | 1,176 | 7,577 | 191 |
| 1924 | Kölnische Zeitung | 141 | 79,444 | 677,256 | 9,059 |
| 1928 | Dortmunder Genera-Anzeiger, Unterhaltungsblatt | 2 | 1,127 | 7,530 | 81 |
| 1929 | Rheinisches Volksblatt, Illustrierte Beilage | 4 | 554 | 3,796 | 125 |
| 1930 | Hildener Rundschau, Illustrierte Beilage | 8 | 726 | 4,617 | 185 |
| 1930 | * Lippische Zeitung, Volksblatt Illustrierte | 4 | 573 | 4,565 | 102 |
| 1933 | Vossische Zeitung | 4 | 1,051 | 8,071 | 154 |
| Sum | | 801 | 371,642 | 3,127,135 | 32,451 |

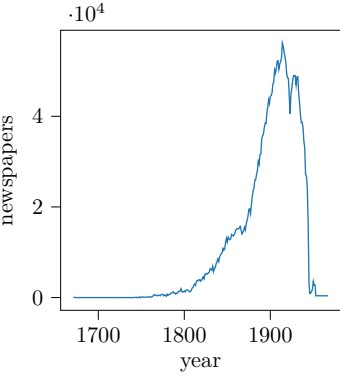 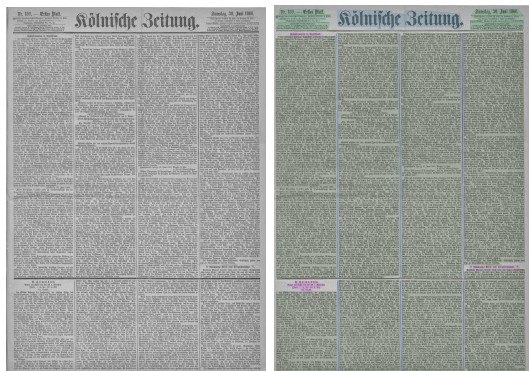

Figure 1: Left: Number of available digitized newspapers per year at `www.deutsche-digitale-bibliothek.de/newspaper` over time. Data from January 2024. Center: Front page of the *Kölnische Zeitung* from the $1^{st}$ of January 1924. Right: Corresponding annotation to the page at the center of this figure.

Table 2: Data-set sizes of this paper and related work in comparison.

| Dataset | Pages |
|---|---|
| Chronicling Germany (ours) | **801** |
| Europeana (Clausner et al., 2015) | 528 |
| News Eye Finnish (Muehlberger and Hackl, 2021c) | 200 |
| Reichs- und Staatsanzeiger (UB-Mannheim, 2023b) | 197 |
| News Eye French (Muehlberger and Hackl, 2021a) | 183 |
| Neue Züricher Zeitung (Ströbel and Clematide, 2019) | 167 |
| News Eye Austrian (Muehlberger and Hackl, 2021b) | 158 |
| News Eye Competition (Michael et al., 2021) | 100 |

errors. Overall, our dataset includes more than 30,000 layout polygon regions as well as more than 370,000 text lines and approx. 3 million words. See Table 1 for an overview of the dataset details.

This dataset is larger than the Europeana corpus Clausner et al. (2015) with its 528 pages from European newspapers, the 197-page "Deutscher Reichsanzeiger und Preußischer Staatsanzeiger" (UB-Mannheim, 2023b) data set or the 167-page Neue Züriche Zeitung (Ströbel and Clematide, 2019; UB-Mannheim, 2023a) corpus. Ground truth pages are compared in Table 2. Our dataset adds a significant number of new pages.

Our dataset features sections and elements that are especially challenging for OCR and baseline models. For example, advertisement pages mix large and small font sizes and include drop capitals, where the initial letter of an advertisement spans over multiple rows but is read as part of the first row. Both features are a challenge for the baseline detection task. Other challenges are fractions in stock exchange news and abbreviations in lists of casualties.[3] Overall, 83,8% of our dataset scans fall into a range between 4500 and 5499 width times 6500 and 7499 height pixels.

In summary, this paper makes the following contributions: (1) We introduce the *Chronicling Germany*-dataset. Its 801 manually annotated high-resolution pages make it the largest German-language historic newspaper dataset to date (see Table 1 for dataset details). (2) We establish a baseline recognition pipeline for the layout detection, text-line recognition, and OCR-tasks. (3) We verify generalization properties using the Out of Distribution (OoD) part of our test set which

---

3. Our dataset includes pages from 1866, when the Austro-Prussian War was raging in the German Bund.

contains 20 pages from five historic newspapers. We observe good generalization performance for OCR and layout tasks. We share processing code as well as the data via online repositories (see below).

- Data: `https://gitlab.uni-bonn.de/digital-history/Chronicling-Germany-Dataset`

- Code: `https://github.com/Digital-History-Bonn/Chronicling-Germany-Code`

## 2 Related Work

Unfortunately, from a digital history perspective, many modern systems focus on recent data and suffer from poor performance in a historical setting.[4] The current situation has led to a large body of OCR error correction work (Carlson et al., 2023), highlighting the need for specialized data sets and software. Liebl and Burghard (2020), for example, combine existing open-source components for this task.

Related datasets include the Europeana corpus (Clausner et al., 2015), the Deutsche Reichsanzeiger (UB-Mannheim, 2023b), and the Neue Züricher Zeitung (UB-Mannheim, 2023a; Ströbel and Clematide, 2019). The Europeana dataset contains *528* annotated pages from European sources. The Reichsanzeiger and the Neue Züricher Zeitung sets consist of *197* and *167* annotated pages, respectively, but have so far only been used for OCR training but not in any layout training pipeline (cf. 2). The layout annotation of these two projects is comparable to ours but less granular, and we have annotated considerably more pages.[5] More recently Dell et al. (2024), published perhaps the largest American historical newspaper dataset to date. Their dataset also includes layout annotations. Our work complements these existing datasets by additionally providing compatible annotations for German historical newspapers that differ significantly from other Western European and American newspapers. Furthermore, we annotate advertisements in detail, which significantly add to the complexity of the OCR-task (Dell et al., 2024) and are not annotated in the Reichsanzeiger and the Neue Züricher Zeitung. Advertisements are particularly interesting to scholars of economic history who are interested in labour markets, for example. Globally, the Historical Japanese Dataset (Shen et al., 2020) , a collection of over 2,000 annotated pages of complex layouts from the Japanese Who is Who of 1953, is notable.

### 2.1 Common processing pipeline elements

**Layout Segmentation**   is a longstanding task in document processing. For example, dhSegment (Oliveira et al., 2018) proposes a UNet structure based on the popular ResNet50 architecture (He et al., 2016). As described by Ronneberger et al. (2015), the network features a contracting and an expanding part. The contracting subnetwork uses ResNet50 as an encoder, and an additional expansive subnetwork produces segmentation maps at the resolution of the original input. Transformer-based solutions trained on modern documents are available for similar tasks (Davis et al., 2022). However, Convolutional Neural Networks (CNNs) are cheaper to run (Dell et al., 2024) and require less training data, making them a budget-friendly solution.

**Baseline-detection**   or text-line detection means finding the straight line that connects the base points from each letter. Early work employed quadratic splines for this task (Smith, 2007). Modern solutions often employ architectures devised for segmentation or object detection tasks. Kodym and Hradis (2021) for example, choose a U-Net. Object detection pipelines are alternatively used instead of baseline detection; e.g., Dell et al. (2024) work with YOLOv8. Following Kodym and Hradis (2021), we employ a U-Net to detect text baselines and use them to generate line polygons. Our annotations are consistent with the Europeana-corpus from Clausner et al. (2015) and the work from

---

4. For an example see Figure 6 in A.3, alco compare Shen et al. (2020).
5. We aim to combine those two datasets with Chronicling Germany in future work.

Table 3: Dataset split (left), test data is divided into in Distribution (iD)- and Out of Distribution (OoD)-data. The right hand side shows label distribution-percentages per pixel.

| | pages | polygons | lines |
|---|---|---|---|
| train | 651 | 27,162 | 324,399 |
| validation | 50 | 1,784 | 15,024 |
| test iD | 80 | 2,851 | 26,396 |
| test OoD | 20 | 654 | 5823 |
| sum | 801 | 32,451 | 371,642 |

| label | class | frequency |
|---|---|---|
| 0 | background | 34.11% |
| 1 | caption | 0.77% |
| 2 | table | 3.10% |
| 3 | paragraph | 58.50% |
| 4 | heading | 1.11% |
| 5 | header | 0.74% |
| 6 | separator vertical | 0.68% |
| 7 | separator horizontal | 0.63% |
| 8 | image | 0.32% |
| 9 | inverted text | 0.02% |

UB-Mannheim (2023a,b) that also features Fraktur letters. This design choice allows us to combine our datasets in future work.

**Optical Character Recognition (OCR)** is an important tool in digital history. Liebl and Burghard (2020), successfully work with a topological feature extraction step followed by a classifier as described by Smith (2007) for the digitization of the *Berliner Börsen Zeitung*. Following Breuel (2007), Kiessling (2022) uses a Recurrent Neural Network (RNN) based system. Dell et al. (2024) apply the contrastive learning approach presented by Carlson et al. (2023). Using a vision encoder, characters are projected into a metric space. The system works because patches containing the same character will cluster together.

## 3 The Chronicling Germany Dataset

Our Dataset contains 801 pages from historic German newspapers, mostly from 1866, specifically from the period of the Austro-Prussian War. Of these 801 pages, 15 pages contain only advertisements with approx. 1,900 individual advertisement blocks. The backbone of the dataset is the *Kölnische Zeitung*, a large regional newspaper from Western Germany. Additionally we include newspaper pages from before and after the German Empire and various German regions (See Table 1). Overall, we consider the dataset to be a very good representation of the different layout styles of historical German newspapers. (For a more detailed description and justification of the composition of the dataset, see A.4).

We split our data into train, validation, and test datasets (Table 3), where the test dataset consists of in Distribution (iD) and Out of Distribution (OoD). OoD pages are taken from five German newspapers that are not present in the training or validation dataset (Table 1). The train and validation data splits only contain iD data.

Polygons placed by our expert human annotators capture the layout for each page.[6] All annotations are stored in PAGE-XML files. The polygons capture different text-region types. Subclasses can exist within these. Each region type has a unique XML tag: `TextRegion`, `SeparatorRegion`, `TableRegion` and `GraphicRegion`. Graphic regions are always assigned the class `image`. Within text regions, we include the following classes: `paragraph, header, heading, caption, inverted_text`. Within table regions, the only possible subclass is `table`. To facilitate correct reading order detection, we introduce the separator subclass `separator_vertical`, and `separator_horizontal`. Vertical separators highlight different columns of a page. Horizontal separators split the page into sections

---

6. The annotation process is documented in A.4, the annotation guidlines are reported in A.5.

and are relevant for the reading order if they span over multiple columns. Otherwise, they are found at the beginning of a new article or between caption or header elements. The header category covers the newspaper's name, which appears at the top of the front pages. To the left and right of the newspaper name, historical newspapers often have smaller blocks with additional information, such as the name of the editor-in-chief, the publication date, or the subscription price. These polygons are annotated as captions. Polygons that cover paragraphs, headlines, and tables are annotated, respectively. See Figure 1 for an annotation sample. Overall, the dataset includes 32,451 polygon regions.

We primarily use a combination of the classes described above to annotate the historic advertisements. We have decided not to introduce new classes to avoid confounding the model's training. This applies, in particular, to the separator classes. Therefore, we use the classes `separator_vertical` and `separator_horizontal` for the annotation of separator regions around individual advertisements. Advertisements tend to use text blocks with bigger fonts. To be consistent with our annotations, we mark these as `heading`. For the same reason, the normal-sized text is annotated as `paragraph`. Additionally, we include the classes `inverted_text` and graphic elements as `image`. These are present, especially in the advertisement pages, as well as in pages from 1924. Table 3 illustrates this numerically. The two classes `inverted_text` and `image` are only present in a subset of the data, which explains its low share of pixels overall.

Regions of each page have a reading order number assigned to them. These numbers are assigned automatically and not corrected manually. Reading order is not the main scope of this dataset. Automatic assignment leads to satisfactory results for most pages. For advertisement pages, however, it does not. Yet, advertisements don't need a meaningful reading order, as they are comprised of elements that are independent of each other.

In addition to the layout data, we include transcribed text divided into text lines. In our dataset, each text line is comprised of a polygon, which contains all characters, as well as a baseline and the corresponding text. Baselines and text transcriptions are generated automatically and then corrected by expert annotators. Partially the generation was done using the pipeline proposed by Kodym and Hradis (2021) , more recent additions to the dataset have been generated using our own pipeline. Line polygons and baselines are only corrected when there are significant mistakes. This is especially the case within the advertisement pages, where some initial letters of advertisements span over more than one line. Correct drop capital detection is challenging for current text-line detectors. Overall, our dataset includes 371,642 text lines. The transcription follows the OCR-D guidelines, level 2 (Johannes Mangei, 2024). This means the text is transcribed in a visual style, preserving, for example, the archaic 'long s' or ' ſ'. For a complete discussion, see supplemental section A.5.

## 4 Experiments and results

**Data:**  All experiments work with fixed train, validation and test splits as outlined in Table 3.

**Pipeline:**  Figure 2 presents a pipeline overview. Overall, we employ two U-Nets for layout recognition and text-line detection and, finally, a Long Short-Term Memory (LSTM) network for OCR. The pixel-wise layout inference is converted into polygons during the post-processing step. We use targets like Kodym and Hradis (2021) for training the baseline U-Net. The model recognizes baselines, ascender-, descender-, and end-points, which are converted into line regions and baselines during post-processing. The post-processing code is an adapted version from Kodym and Hradis (2021). Contrary to their approach, we use the layout regions from the previous step to cut out parts of the image and identify all baselines for each region. These baselines are then used as input for the LSTM OCR model and the original image.

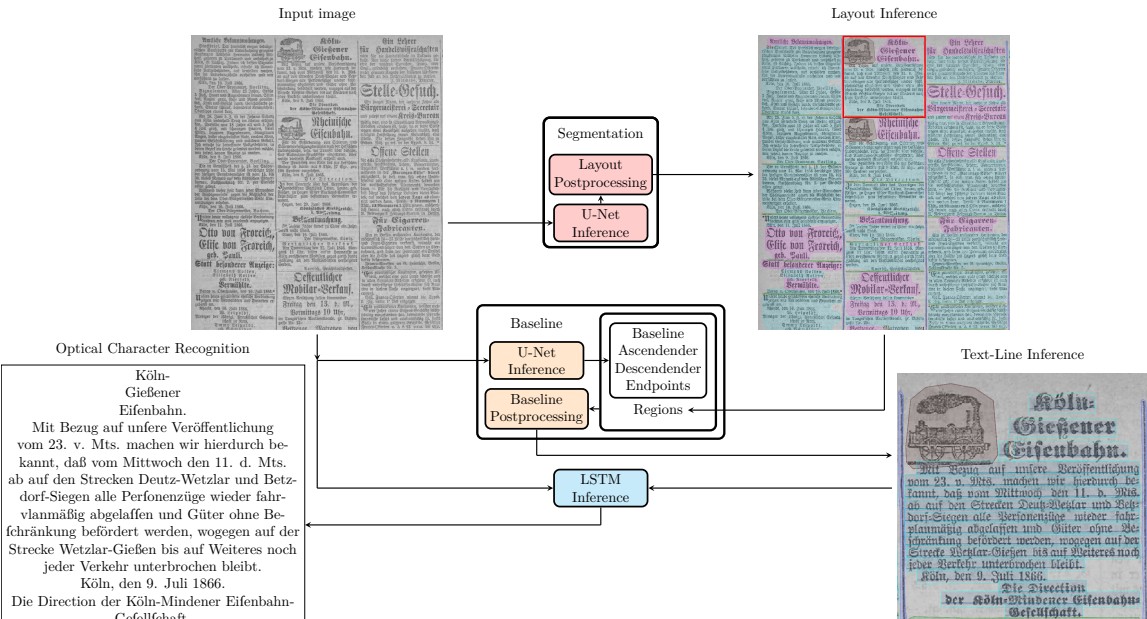

Figure 2: Flow chart of the entire prediction pipeline. The layout detection, text-line inference and Optical Character Recognition (OCR)-tasks use separate networks each. The output is machine-readable and can be processed further. For example in a machine translation step.

## 4.1 Layout-Segmentation

**Training:** Our layout segmentation setup follows Oliveira et al. (2018). For layout training, all pages are scaled down by a factor of 0.5 and split into 512 by 512-pixel crops. Cropping leads to 39,072 training crops overall. During training, we work with 32 crops per batch per graphics card. The training runs on a node with four graphics processing units (GPUs). Consequently, the effective batch size is 128, with 305 training steps per epoch. Initially, optimization of the contracting network part can start from pre-trained ImageNet weights, while optimization of the expanding path has to start from scratch. The expanding subnetwork starts with the encoding from the contracting network and produces a segmentation output at the input resolution. To improve generalization, input crops are augmented using rotation, mirroring, Gaussian blurring, and randomly erasing rectangular regions. An AdamW-Optimizer (Loshchilov and Hutter, 2017) trains this network with a learning rate of 0.0001, with a weight decay parameter of 0.001 for 50 Epochs in total, while using early stopping to save the best model. We initialize the ResNet-50 encoder using ImagNet weights before training. We use only in distribution data for training and validation.

**Results:** Table 4 column two lists network performance on the test dataset, and column four lists performance on the id test data only. We compute F1 Score values for all individual classes on pixel level. Generally, we find good performance with both datasets. Table 10 additionally shows results for OoD only data. Overall, the results show that rare classes, especially `image` and `inverted_text`, are a challenge, while the `paragraph` class shows very good performance and generalization. Most noticeably, our pipeline displays much better performance for the `caption` and `header` classes for iD in contrast to OoD data (Table 9). We suspect that this behavior is caused by specific caption patterns. In OoD data, the caption pattern is often unusual compared to iD data, which creates this

Table 4: Layout detection test set results. This table lists F1 Score values for all individual classes. N/A values in columns (3) and (5) are due to differing annotations between our approach and Dell et al. (2024). In columns (2) and (4) we report our model results for in Distribution (iD) + Out of Distribution (OoD) and in Distribution (iD) only data. (Out of distribution only results for layout in Table 10)

| | F1 Score (iD + OoD) | | F1 Score (iD only) | |
|---|---|---|---|---|
| class | ours | Dell et al. | ours | Dell et al. |
| background | $0.96 \pm 0.01$ | 0.79 | $0.93 \pm 0.03$ | 0.79 |
| caption | $0.63 \pm 0.10$ | N/A | $0.85 \pm 0.01$ | N/A |
| table | $0.90 \pm 0.02$ | 0.43 | $0.90 \pm 0.03$ | 0.45 |
| paragraph | $0.97 \pm 0.01$ | 0.88 | $0.95 \pm 0.03$ | 0.89 |
| heading | $0.67 \pm 0.04$ | 0.46 | $0.69 \pm 0.03$ | 0.46 |
| header | $0.80 \pm 0.03$ | 0.23 | $0.80 \pm 0.05$ | 0.26 |
| separator vertical | $0.72 \pm 0.02$ | N/A | $0.72 \pm 0.03$ | N/A |
| separator horizontal | $0.74 \pm 0.01$ | N/A | $0.73 \pm 0.02$ | N/A |
| image | $0.70 \pm 0.06$ | N/A | $0.74 \pm 0.06$ | N/A |
| inverted text | $0.16 \pm 0.02$ | N/A | $0.27 \pm 0.14$ | N/A |

effect. Figure 4 and Figure 5 present newspaper pages from our test set with layout ground truth and prediction samples side by side.

We also compare our model results to the pipeline developed by Dell et al. (2024). To his end, we run their pipeline on our test data. Columns three and five of table 4 list the results. Overall, our pipeline performs better on the comparable classes of our test dataset than Dell et al. (2024). However, we do not fine-tune their model on our training data, and there are significant differences between the Chronicling Germany dataset and the American Stories dataset that distort the comparison. Dell et al. work with a YOLOv8 object detection model for layout recognition. We have not been able to fine-tune their pipeline. Instead, we evaluate another YOLOv8 model that has been fine-tuned on our dataset. Table 11 shows full results of the fine-tuned YOLOv8. The most significant difference between Dell et al. and our data is the more detailed headline annotation of the Chronicling Germany dataset. [7] Dell et al. (2024) seems to assign the headline class less frequently to headlines that do not stand out clearly. This leads to a significant amount of headline regions from the Chronicling Germany test-set to be classified as paragraphs by the Dell-pipeline. Furthermore, the Chronicling Germany dataset includes annotated separator regions, while American Stories does not.[8] Moreover, Dell et al. (2024) treats the entire header of a newspaper front page as one class, while Chronicling Germany differentiates between the header itself and the captions that are typically left and right of the header.

## 4.2 Baseline Detection

**Training**: Following Kodym and Hradis (2021) we train an U-Net for the text-baseline prediction task. The raw input image as well as ground truth baselines serve as starting points for the optimization. The training process minimizes a joint text-line and text-block detection objective as introduced by Kodym and Hradis (2021). We run an AdamW-optimizer (Loshchilov and Hutter, 2017) with a

---

7. Segmentation of headlines is of particular interest for historical research, as they allow for the layout-based identification of and differentiation between individual articles.
8. In the case of the Kölnische Zeitung, identifying horizontal page-spanning separators is of crucial importance for the reading order, as they act like a page break. This means the reader should not continue reading down the current column but go back up to the next one.

Table 5: Baseline detection test set results. We measure performance in precision, recall and F1 score. Detected lines are matched with ground truth lines and are considered a true positive if the predicted line has an IoU score of more than 0.7 compared to the corresponding ground truth line. Results are averaged over all test pages.

| Model | precision | recall | F1 score |
|-------|-----------|--------|----------|
| UNet | $0.938 \pm 0.007$ | $0.897 \pm 0.013$ | $0.917 \pm 0.010$ |

learning rate of 0.0001 and a batch size of 16. During training, inputs are randomly cropped to 256 by 256 images. To improve the robustness of the resulting network, the input pipeline includes color jitter, Gaussian blur, random grayscale, random scaling, and Gaussian blur perturbations during training.

**Results:** We measure precision, recall, and F1 score (see Table 5 ). Generally, we observe values around 0.9. These observations are in line with Kodym and Hradis (2021), who observe similar numbers on the cBAD2019 dataset (Diem et al., 2017). Dell et al. (2024) do not provide baseline data, instead opting for extracting the text for each region as a whole employing a Yolov8 architecture. Since we do not annotate the ground-truth text boxes, there is no adequate way to compare the two pipelines for this specific task. The historic newspaper community either works with baseline detection or direct text object detection pipelines. We decided to follow Kodym and Hradis (2021) and employ a U-Net to detect text baselines. This is consistent with other European projects like the Europeana-corpus from Clausner et al. (2015) and the work from UB-Mannheim that also features Fraktur letters. This choice allows combining these European datasets in future work, and is a key design choice, since we aim to boost performance in the Fraktur-subset of historical newspapers.

### 4.3 Optical Character Recognition (OCR)

**Training:** Based on the Kraken-OCR-engine (Kiessling, 2022) we train a LSTM-cell for the OCR -task and employ baselines to extract individual line polygons. Alongside the annotations, which our human domain experts have checked, these serve as input and ground truth pairs. Adam (Kingma and Ba, 2015) optimizes the network with a learning rate of 0.001. Optimization runs for eight epochs with a batch size of 32 sequences. We use early stopping to prevent the models from overfitting and include pixel-dropout, blur, rotation, and see-through-like augmentations during training to improve generalization. Additionally, we train and evaluate the OCR-transformer proposed by Kodym and Hradis (2021) as part of their pero-application with AdamW optimizer (Loshchilov and Hutter, 2017) and a learning rate of 0.0001, using the same augmentations.

**Results:** Compared to the model trained by the Universitätsbibliothek Mannheim (Jan Kamlah, 2024), we observe slightly improved performance when finetuning their model as well as training a new model from scratch (Table 6). Overall, we achieve a Levenshtein distance of 0.022 per character, which corresponds to 97.8% of all characters being correctly recognized. We will refer to this character recognition rate as the score.

The corresponding model achieves a perfect score on 68.2% of all text lines, and 96% of all text lines achieve a score of at least 90%. (see: Table 6 ). Overall, our OCR-transformer achieves a score similar to the original (Jan Kamlah, 2024) Kraken model. Our LSTM cell performs slightly better. We notice, however, that the OCR-transformer produces the worst results on OoD test data. At the same time, it performs best on iD data. We suppose that this is a sign of overfitting on the iD data. The Antiqua-pretrained OCR model from Dell et al. (2024) does not generalize well to the Fraktur

Table 6: Optical Character Recognition (OCR) test set results. Levenshtein distance per character appears in the first column. We compute the percentage of completely error-free lines for each model. The second column lists these results. Finally, we consider a line to have many errors if we observe a Levenshtein distance of more than 0.1 per character. We report the percentage of many error lines in the final column. We list the mean and standard deviation for multiple seeds.

| Model | Data | Levenshtein-Distance | fully correct [%] | many errors [%] |
|---|---|---|---|---|
| LSTM (UBM 2024) | iD + OoD | 0.029 | 47.1 | 6.1 |
| | iD only | 0.029 | 48.5 | 6.0 |
| | OoD only | 0.030 | 40.8 | 6.4 |
| LSTM finetuned (ours) | iD + OoD | $0.025 \pm 0.002$ | $67.3 \pm 0.6$ | $4.3 \pm 0.2$ |
| | iD only | $0.026 \pm 0.002$ | $70.1 \pm 0.5$ | $4.4 \pm 0.2$ |
| | OoD only | $0.018 \pm 0.001$ | $59.2 \pm 1.5$ | $4.1 \pm 0.3$ |
| LSTM (ours) | iD + OoD | $0.022 \pm 0.002$ | $68.2 \pm 0.7$ | $4.0 \pm 0.2$ |
| | iD only | $0.023 \pm 0.002$ | $70.4 \pm 0.6$ | $4.0 \pm 0.3$ |
| | OoD only | $0.019 \pm 0.001$ | $58.4 \pm 1.7$ | $4.1 \pm 0.2$ |
| Transformer (ours) | iD + OoD | $0.028 \pm 0.001$ | $64.1 \pm 0.4$ | $8.6 \pm 0.3$ |
| | iD only | $0.021 \pm 0.001$ | $69.5 \pm 0.3$ | $5.3 \pm 0.2$ |
| | OoD only | $0.065 \pm 0.003$ | $39.5 \pm 0.8$ | $23.7 \pm 1.0$ |

texts. For their pipeline, we observe an average Levenshtein distance of 0.58 on the test set (not included in Table 6).[9]

## 4.4 Overall pipeline performance

So far, we have evaluated all components individually using ground truth inputs from previous steps. Next, we additionally run the complete pipeline on the test set (Table 7). We choose the best model for each component, according to the validation set (50 pages), to use in our pipeline. Then, we evaluate the resulting transcription with our ground truth. All predicted and ground truth lines are matched based on the intersection over union of the corresponding text lines. Lines without a match are paired with an empty string. Our pipeline achieves an overall Levenshtein distance per character of 0.07 (93% score) across the entire test dataset. The significant difference to the OCR-only evaluation in Table 6 is due to errors in the layout prediction step. These errors cause prediction text lines to not be matchable with the ground truth lines. Table 8 compares results of the original pipeline with a modified version, where different text regions are merged and unmatched lines are ignored. The modified pipeline evaluation results in a Levenshtein distance of 0.032 (96.8% score), similar to the OCR-only evaluation (Table 6).

## 5 Pipeline-Generalization

**Test-Data:** To verify generalization, our test dataset contains 20 Out of Distribution (OoD) pages from different papers and time periods (Table 1 and Table 3).

**Inference:** In addition to the full test set evaluation, we run the entire pipeline on the OoD data only (Table 7). We measure a Levenshtein distance per character of 0.08 (92% score). When ignoring

---

9. Please note that we cannot fine-tune the OCR engine proposed by Dell et al. (2024) on our Fraktur data because of differences in the text detection step (see above).

Anzeige
für Kunftreite
In dem fchönen und fehr befuchten Park in Stock-
holm ift der neuerbaute Circus vom Anfang nächften
Mai-Monats an zu vermiethen. Das Local, welches
mit den großten und eleganteften diefer Art in Europa
wetteifert, ift mit einem befonders fchon eingerichteten
Schauplatz verfehen, auf welchem mimifche und dra-
matifche Vorftellungen gegeben werden konnen. Es
find gleichfalls bequeme Wohnzimmer für die Gefell-
fchaft zu haben. Außer der Bühne find wohleinge-
richtete Stalle vorhanden. Reflectirende können über
die Miethebedingungen mit dem Befitzer, dem Ma-
fchinenmeifter Hrn. C. Mothender, Blafiebolmen
No. 4. Stockholm, in Correfpondenz treten. Nähere
Auskunft ift gleichfalls bei Hrn. Joh. Hüllmann
ltona

Figure 3: Generalization test set sample image. This figure shows a page element with detected baselines on the left. The right side presents the automatically created transcription.

Table 7: Optical Character Recognition (OCR) test set results for running the entire pipeline on our dataset. Levenshtein distance per character appears in the first column. We compute the percentage of completely error-free lines for each model. The second column lists these results. Finally, we consider a line to have many errors if we observe a Levenshtein distance of more than 0.1 per character. We report the percentage of many error lines in the final column. All predicted and ground truth lines are matched based on the intersection over the minimum of the corresponding text lines.

| Model | Data | Levenshtein-Distance | fully correct [%] | many errors [%] |
|---|---|---|---|---|
| | iD + OoD | 0.070 | 53.4 | 21.1 |
| Full Pipeline | iD only | 0.065 | 55.7 | 19.7 |
| | OoD only | 0.080 | 43.3 | 27.6 |

unmatched lines and merging different text regions into one label, the results significantly improve to 0.038 (96.2% score) (Table 8). Figure 3 presents an OoD example taken from a 1849 issue of the Hamburger Correspondent. The sample is an advertisement for a circus in Stockholm that is available for rent. The advertisement describes the quarters and stables which belong to the circus. It includes the name and address of the owner, to allow interested parties to reach out for additional information. The names and addresses contain several abbreviations, which are a challenge for OCR. In this example, most abbreviations are recognized correctly. Linguistically, the sample is close enough to modern German to be machine-translated.

## 6 Conclusion and future work

This work introduces the *Chronicling Germany* -dataset, the currently largest dataset of German language historical newspaper pages. In addition to the dataset, it presents a neural network-based processing baseline with test-set OCR-accuracy results. Our paper creates a starting point for researchers who wish to improve historical newspaper processing pipelines or are looking for a low-resource computer vision challenge. To create the dataset, history students spent 2,700 hours annotating the layout. The dataset's 801 pages make it the largest fully annotated collection of historic German newspaper pages to date. The dataset includes 1,900 individually annotated advertisements. Furthermore, the Out of Distribution (OoD) part of our test set includes 20 pages selected from historic newspapers that are not part of the training or validation set. We verify baseline pipeline performance on the OoD pages. By following the OCR-D annotation guidelines (Johannes Mangei, 2024), we ensure our annotations' compatibility with concurrent and future work.

**Acknowledgments**

We thank Jan Göttfert, Charlotte Lüttschwager, and Ida Wenzel for helping annotate the data. We furthermore want to thank Paul Asmuth, Judith Eifler, Daniel Göttlich and Moritz von Kolzenberg, and Robin Weiden for supporting the project.

This project was funded by the University of Bonn's TRA1 (Mathematics, Modelling, and Simulation of Complex Systems) and TRA4 (Individuals, Institutions, and Societies) as part of the Excellence Strategy of the German federal and state governments. In particular, we would like to extend our gratitude to Dr. Daniel Minge and Johanna Tix for helping fund our dataset collection efforts. Furthermore, research was supported by the Bundesministerium für Bildung und Forschung (BMBF) via its "BNTrAInee" (16DHBK1022) and "WestAI" (01IS22094A) projects.

The authors gratefully acknowledge the Gauss Centre for Supercomputing e.V. (www.gauss-centre.eu) for funding this project by providing computing time through the John von Neumann Institute for Computing (NIC) on the GCS Supercomputer JUWELS at Jülich Supercomputing Centre (JSC).

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

# Appendix A. Supplementary Material

## A.1 Acronyms

**CNN** Convolutional Neural Network

**GPU** graphics processing unit

**iD** in Distribution

**LSTM** Long Short-Term Memory

**NLP** Natural Language Processing

**OCR** Optical Character Recognition

**OoD** Out of Distribution

**RNN** Recurrent Neural Network

## A.2 Machine learning is important for the study of history

Figure 1 illustrates the breakthrough of the newspaper industry in the 19th century.[10] While the number of newspapers listed in the *Deutsche Zeitungsportal* grew at a rate of 2.9 percent p.a. in the first two-thirds of the 19th century, the increase rose to 3.4 percent p.a. after the foundation of the German Empire. There were three main reasons for this increase: firstly, the literacy of the population increased over the century. Secondly, considerable technological advances made it easier to produce a newspaper. Thirdly, state control of newspapers declined from the middle of the century. A significant milestone was the Press Act of 1874, which finally abolished censorship (although some restrictions remained so that even after 1874, there was no complete freedom of the press in the German Empire). Nevertheless, no later than the last third of the 19th century, a mass market for print media had emerged in Germany, which was served by many newspapers whose content and political orientation were very heterogeneous.

Historical newspapers contain a wealth of information about past societies. They provide information about the spatial occurrence of events, about contemporary perceptions of social and economic change, and allow tracing of cultural change. Blevins (2014), for example, uses the mentioning of place names in the *Houston Post* to draw a mental map of the Nation around 1900. Measured by the mention of place names, the region west of Houston was deeply rooted in the newspaper and its readership. The East Coast and the Midwest were also present in the imagination of contemporaries. However, the Southwest, the Northwest, and California hardly appear on this mental map. Based on the newspaper's coverage, one could argue that readers of the *Houston Post* around 1900 were barely aware of the Nation as a geographical entity. In economic history, historical newspapers have recently been used to identify treatments or measure variables of interest. Beach and Hanlon (2023) give an overview of the recent use of historical newspaper data in economic history. An interesting recent example is Ferrara et al. (2024), who used digitized newspaper archives to measure a county's exposure to the boll weevil around 1900. The boll weevil is a pest of cotton that hit the American South between 1892 to 1922. The pest reduced cotton production and, consequently, hastened social changes in the primarily Black rural communities, like the fertility transition and higher investment in education.

Even though newspaper portals are an essential source for historians and other disciplines interested in history, such as economics, their potential has not yet been fully realized (Beach and Hanlon, 2023) . Firstly, researchers have so far mainly used US-American portals. The reason for this

---

10. Note that the Deutsche Zeitungsportal does not collect all historical newspapers. There is probably a selection bias towards more prominent outlets with extended publication periods. However, on the whole, figure 1 should reflect the development of the newspaper market in Germany quite well.

bias may be these portals have been established longer than in other regions of the world. Secondly, the mass utilization of newspaper data is often limited to a keyword search, which usually only covers the entire page and does not discriminate between articles. Therefore, the joint occurrence of two or more search terms is recorded for the page, not the article, and information retrieval is thus still very imprecise (Oberbichler and Pfanzelter, 2021). Thirdly, the text cannot always be downloaded easily, which makes further processing by researchers more difficult. On the other hand, the image files of individual newspaper pages are easy to obtain via the portals. Deep learning algorithms that recognize the layout of a newspaper page and capture the text at the article level, therefore, promise great benefits for historical research. The *Chronicling Germany* data set presented here, comes with layout annotations for every page. It is intended to stimulate the further development of deep learning algorithms and to promote the increased use of non-American newspaper portals.

In addition to more accurate and straightforward information retrieval, downloadable article-level data will also allow scholars of history to apply advanced NLP-methods in the future, including document and text embedding techniques and fine-tuning large language models to 19th-century German.

## A.3 Additional figures

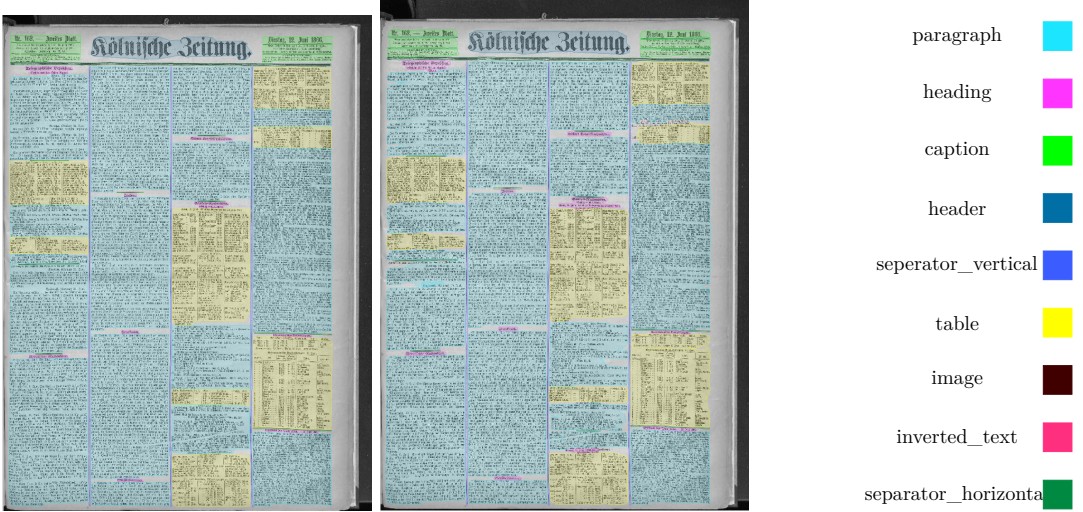

Figure 4: Target labels on the left and segmentation prediction on the right. Newspaper page of the Kölnsche Zeitung from the iD test set.

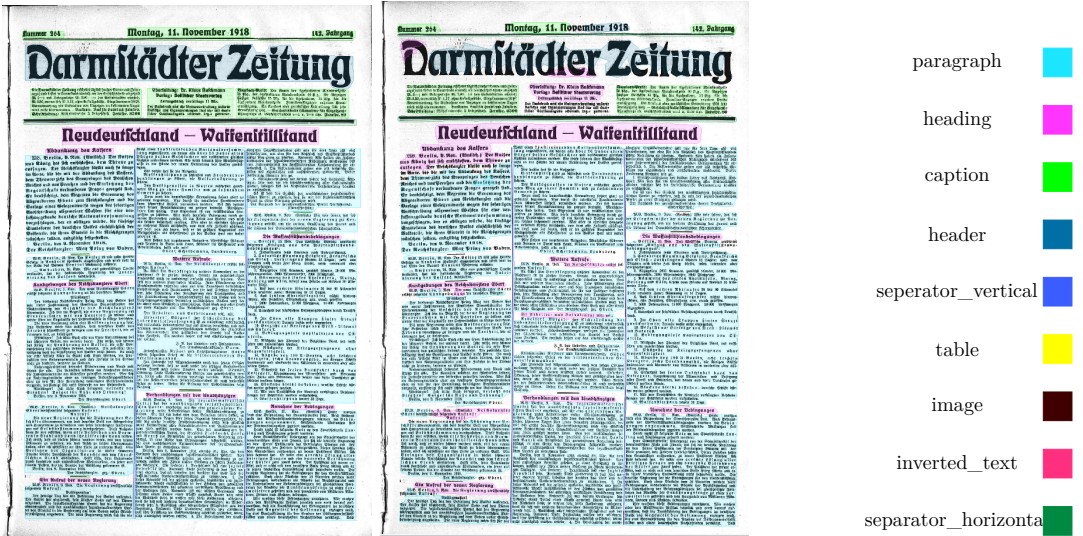

Figure 5: Target labels on the left and segmentation prediction on the right. Newspaper page of the Darmstädter Zeitung from the OoD test set.

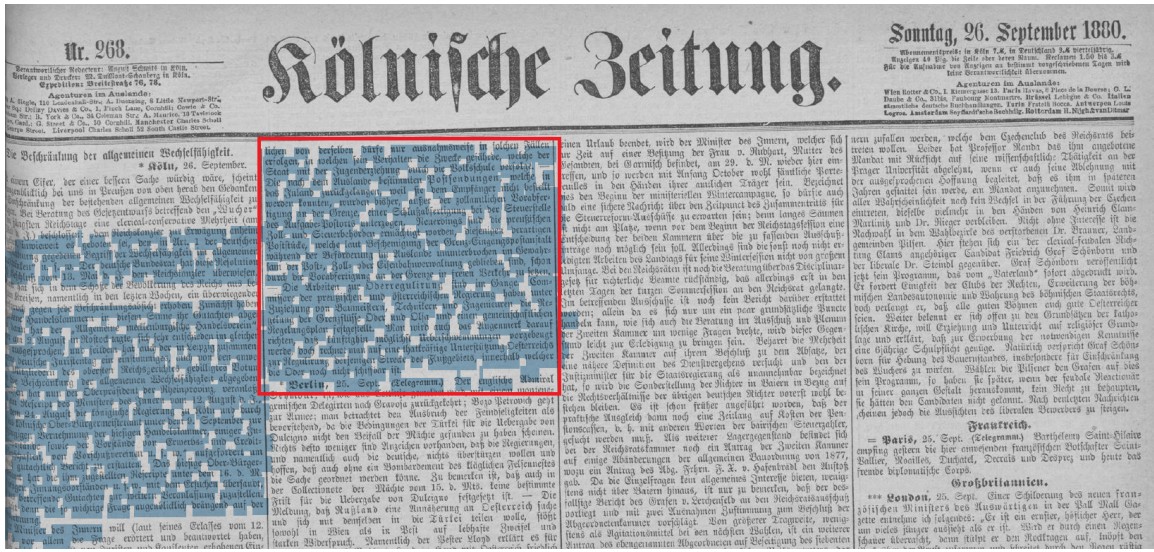

Figure 6: Layout recognition error in the *Kölnische Zeitung*. A researcher tried to select text in the first column on the very left but the column layout is not understood correctly. This page was published on September 26, 1880. Digital versions are available at the *zeit.punkt NRW* website. Layout recognition and transcription generated by Transkribus.

## A.4 Dataset description

### A.4.1 Construction of the dataset

The Chronicling Germany dataset includes 30 newspapers from all over Germany (Figure: 7). Most of the data (581 pages) is from the Kölnische Zeitung but we have added 220 pages from other newspapers covering the time period 1617-1933 (Table 1). Currently, these 30 newspapers are completely in the test set. However, in an updated version we will also include different newspapers into the training pipeline. Overall, we have annotated 801 pages including over 3 million words and ca. 32,000 region polygons. Aside from availability and representation, we selected these newspapers for the following reasons: Our focus is on 1866, the year of the Austro-Prussian War. Aside from its historical importance as the second of the three unification wars and the decisive turning point towards the "Kleindeutsche Lösung", this year gives certain advantages to make our data more diverse: During this year, most newspapers across Germany reported lists of the fallen, missing and deserted as well as reports on military careers of officers. These are usually printed in a considerably different layout, thus diversifying our data. Additionally, most states - particularly during wartime - obliged newspapers in their territory to publish "Öffentliche Bekanntmachungen" or official notices. In 1866, all states handled this differently, resulting in more diverse newspaper layouts across Germany (compared to 1871). Also , focusing on this year allows users of our dataset to evaluate separately how well a model generalizes to different newspapers of the same time and how well it generalizes to newspapers from other decades. When no newspapers from 1866 were available, we sometimes included an issue from 1867. The different newspapers have been chosen to maximize variation between them. We include newspapers from various regions of (past) Germany, like Berlin, Eastern Prussia, the Rhineland, Lower Germany (Hannover), Bavaria, The Palatinate, and Saxony. We also take care to include larger national as well as regional newspapers, and newspapers with a special non-political interest, like the Neue Berliner Musikzeitung (New Berlin Musics Paper) and Der Bazar (a paper on "women's topics" - mostly written by men). We dedicated extra attention to the Vossische Zeitung, because it is one of the most-read newspapers of its age and - due to its bad printing quality - it is particularly difficult for layout detection and OCR. The amount of pages per newspaper varies, since we include full issues of each newspaper, regardless of their length. This is done to ensure that the entire diversity in layout and font across different sections of the newspaper is represented in the dataset. The years from 1924 onwards constitutes a natural end for the dataset, since German newspapers gradually started using latina fonts instead of Fraktur during that period.

### A.4.2 Annotation process

The annotation process follows the annotation guidlenes in A.5. A human domain expert carried out all layout annotations that were then cross-checked by another human domain expert. Annotating and correcting text is extremely time-consuming. For the moment an automatic transcription is checked and corrected by a single human domain expert. Currently, we have corrected 446 pages in of the Kölnische Zeitung, and 112 pages in the generalization part of the dataset. To improve quality further we will run a second correction round, where all lines will again by proofread by different annotators.

## A.5 Annotation Guidelines

### A.5.1 Introduction

These annotation guidelines are an adaptation of the OCR-D rules ( `https://ocr-d.de/en/gt-guidelines/trans/transkription.html` ). We outline additional rules, we created to ensure consistency of the *Chronicling Germany* dataset.

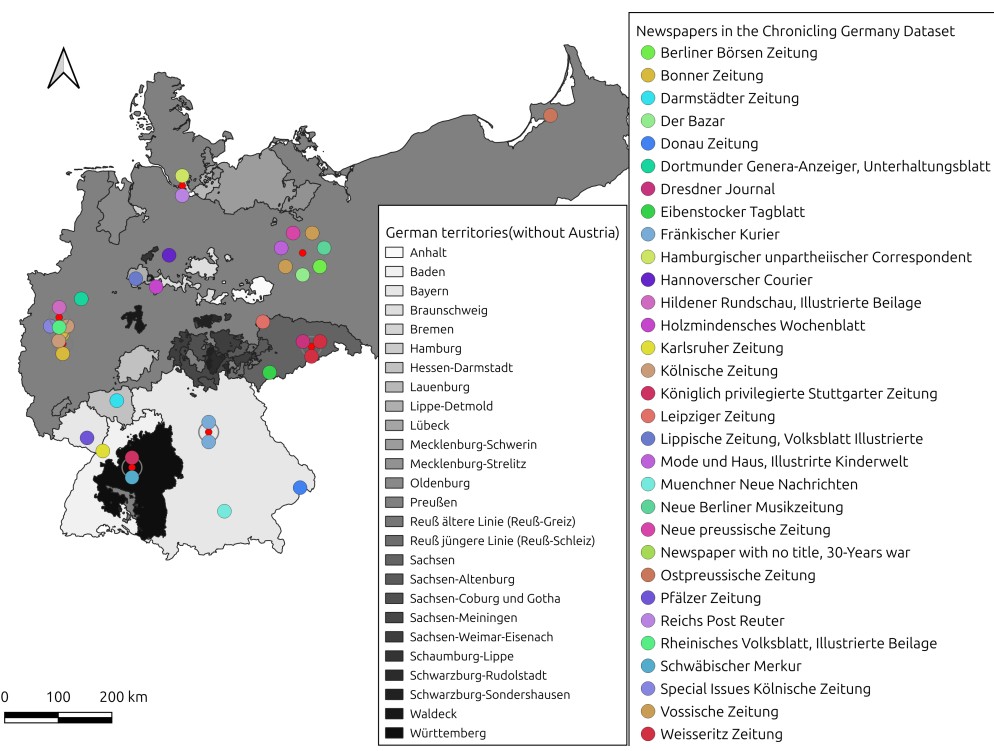

Figure 7: Map of historic Germany from 1867 with labeled regions and regions of newspaper origins beyond the Kölnische Zeitung.

### A.5.2 Page types and type area

The OCR-D guidelines provide for a distinction to be made between page types and the type area during layout analysis. The type area usually contains the text body, but not elements such as the page number. In the *Chronicling Germany* data set, these steps are currently not taken into account.

### A.5.3 Regions

**Region-types** The OCR-D guidelines distinguish between different types of regions, such as text, image and separator regions. In the Bonn Newspaper dataset, the regions are generally recorded in accordance with OCR-D page region level 1 ( `https://ocr-d.de/de/gt-guidelines/trans/ly_level_1_5.html` ). However, tables are also recorded as a separate region and no distinction is made between images and drawings; instead, all images, photos, illustrations and drawings are grouped together under the GraphicRegion. The entire contiguous region is always marked as a block. For text regions, this applies to contiguous blocks of the same class.

- `TextRegion:` All texts that are not tables. Table headings are not marked as a text region.

- `TableRegion:` All parts of the page that contain tabular information. These are often, but not always, clearly recognizable as tables by small separators. Text that is only separated by separators does not count as a table, but a structure must be recognizable that assigns certain meanings to rows and columns. Table headings are included with corresponding tables.

- `SeparatorRegion:` All dividing lines are marked as SeparatorRegion. This also includes decorative elements that, like other separator lines, separate areas from each other and are not purely cosmetic in nature. The separators are divided into vertical and horizontal separators and marked with "separator_vertical" and "separator_ horizontal".

- `GraphicRegion:` All graphics, images, photos, illustrations, and drawings.

### A.5.4 TextRegion subtypes

TextRegions are divided into different subtypes. The subdivision corresponds to the OCR-D guideline for text regions (`https://ocr-d.de/de/gt-guidelines/trans/lytextregion.html#textregionen__textregion_` ). However, drop capitals are treated differently from OCR-D. These are counted as part of the paragraph instead of being marked as a separate text region so that models trained on this data will include them in the correct position in their text output. In addition, headlines (caption) and inverted text (inverted-text) are also recorded in the *Chronicling Germany* data set. Instead of annotating advertisements separately, the classes created for other newspaper pages are applied to the advertisements as far as possible. Because headlines should be visually identified, this leads to a large number of text in the advertisements marked as headlines, which contradicts a semantic definition of a headline. Therefore, it makes sense to treat these pages separately in practice and not differentiate between headings and other text. The following elements from the OCR-D guidelines are not represented in the *Chronicling Germany* dataset due to lack of occurrence: page-number, marginalia, footnote, signature-mark, catch-word, floating, TOC-entry

We discuss the definition for the text subclasses below:

- **paragraph:** Standard text type that includes paragraphs. These are usually kept compact to accommodate as much text as possible in the available space. If a text region cannot be assigned to any other type, it falls under the paragraph label.

- **heading:** Headings that can be clearly distinguished visually from the rest of the text. This is achieved by using a significantly larger or bold font and centered text, which is clearly different from the block layout of paragraphs. A heading is located above a paragraph and is sometimes

separated from the previous text by a separator. A thin separator between the heading and the text can occur. However, if there is too much space between them or a thick separator, the two texts no longer count as belonging together in the sense of heading and paragraph. If a text is not superordinate to a paragraph, it cannot be a heading.

- **header:** Page or column titles that appear prominently above the entire page. These are centered at the top of the page and can appear in different font sizes.

- **caption:** Title lines that are located to the right and left of a page heading or text heading. They often contain information such as the date.

- **inverted-text:** Text that is printed white on black. This is often part of decorative elements but is not marked as a graphic element.

### A.5.5 OCR

A prerequisite for text recognition is baseline or text-line recognition. Both the baseline itself and a polygon around the text line are annotated. These are generated automatically and only corrected if the baseline connects non-contiguous text passages. Lines that have been divided into two baselines are not corrected. Tables and inverted text are not given baselines.

The text is corrected according to its optical appearance. What is written on the page is transcribed, even if there are errors in the print or scan. Completely illegible passages are not transcribed, passage that are largly illegible are transcribed but marked with the "unclear" tag of Transkibus. These passage are not used in training.

Zodiac signs and geometric signs at the beginning of some paragraphs – a peculiarity of the Kölnische Zeitung – are not transcribed.

The transcription is carried out according to level 2 of the OCR-D guidelines ( `https://ocr-d.de/en/gt-guidelines/trans/level_2_2.html` ). This includes the transcription of special characters such as the 'long s' (U+017F) or long hyphens (U+2014, em dash). Consistency with the rest of the data is important here. As these were generated automatically, it is best to look for another example and adopt that version if the special characters are unclear.

Unlike in the OCR-D guideline, fractions are not transcribed with special characters. Instead, the fraction is represented with a slash:
$1\frac{3}{4} = 1\ 3/4$.
In this case, it is important to separate the whole number from the fraction with a space. The same applies to times with an underscore. Example for clock times: $11_{45} =$11_45 or $11._{45} =$11._ 45. (For both, use non-breaking spaces in future (U+202F))

The case of a number that is followed by a unit (e.g. 100M) is not dissolved in the OCR-D guidelines. We always add a space between the number and the unit (e.g. 100M becomes: 100 M).

The units of the Lübische Mark currency are transcribed as follows: the Lübische Mark as Ml., the Lübische Pfennig as dl. and the Lübische Schilling as ßl., where 1 Ml. = 16 ßl. = 192 dl. (`https://www.hagen-bobzin.de/hobby/muenzverein_wendisch.html`).

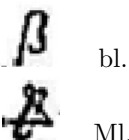
bl.

Ml.

Transkribus allows the selection of special characters with a virtual keyboard. However, it must be ensured that the character used is unique. For example, U+2014 and U+2015 are visually indistinguishable. U+2014 must be used for long hyphens. If the characters are unclear, the OCR-D guidelines, which include tables for the use of special characters, can also be consulted:

- `https://ocr-d.de/en/gt-guidelines/trans/trLigaturen2.html`
- `https://ocr-d.de/en/gt-guidelines/trans/trFremdsprache.html`
- `https://ocr-d.de/en/gt-guidelines/trans/ocr_d_koordinationsgremium_codierung.html`
- `https://ocr-d.de/en/gt-guidelines/trans/trBeispiele.html`
- `https://ocr-d.de/en/gt-guidelines/trans/tr_level_1_3.html`
- `https://ocr-d.de/en/gt-guidelines/trans/trAnfZeichen.html`
- `https://ocr-d.de/en/gt-guidelines/trans/trGedankenstrich.html`

## A.6 Further Results

### A.6.1 Modified Pipeline Results

Table 8: Optical Character Recognition (OCR) test set results for running the entire pipeline on our dataset. We compare the original pipeline with a modified version, where we apply layout merging that combines different text regions and therefore labels all text regions as paragraph. All predicted and ground truth lines are matched based on intersection over union. The modified pipeline is evaluated in a way that ignores unmatched text lines.

| Model | Data | Levenshtein-Distance | fully correct [%] | many errors [%] |
|---|---|---|---|---|
| Original | iD + OoD | 0.070 | 53.4 | 21.1 |
| | iD only | 0.065 | 55.7 | 19.7 |
| | OoD only | 0.080 | 43.3 | 27.6 |
| Modified | iD + OoD | 0.032 | 63.4 | 6.8 |
| | iD only | 0.027 | 65.2 | 6.1 |
| | OoD only | 0.038 | 56.9 | 7.9 |

### A.6.2 Additional Layout Results

Table 9: Full layout detection test set results for our pipeline. This table lists F1 Score values for all individual classes. As the OoD data does not contain the `inverted_text` class, it can not be evaluated on OoD data.

| class | F1 Score (iD + OoD) | F1 Score (iD only) | F1 Score (OoD only) |
|---|---|---|---|
| background | 0.96 ± 0.004 | 0.93 ± 0.030 | 0.89 ± 0.022 |
| caption | 0.63 ± 0.097 | 0.85 ± 0.010 | 0.39 ± 0.132 |
| table | 0.90 ± 0.016 | 0.90 ± 0.027 | 0.67 ± 0.036 |
| paragraph | 0.97 ± 0.005 | 0.95 ± 0.027 | 0.90 ± 0.025 |
| heading | 0.67 ± 0.037 | 0.69 ± 0.025 | 0.50 ± 0.052 |
| header | 0.80 ± 0.025 | 0.80 ± 0.047 | 0.40 ± 0.085 |
| separator vertical | 0.72 ± 0.024 | 0.72 ± 0.027 | 0.58 ± 0.028 |
| separator horizontal | 0.74 ± 0.014 | 0.73 ± 0.016 | 0.57 ± 0.043 |
| image | 0.70 ± 0.064 | 0.74 ± 0.063 | 0.38 ± 0.121 |
| inverted text | 0.16 ± 0.017 | 0.27 ± 0.136 | N/A |

## A.7 Project Limitations and Future Work

### A.7.1 OCR

This dataset contains newspaper pages set in Fraktur-letters. The font is very different from modern fonts. The 'long s' or 'ſ', for example, is completely foreign to modern eyes. While our generalization dataset also includes four pages in Antiqua font, which have been predicted with sufficient accuracy, networks trained exclusively on our dataset are not likely to outperform more specialized networks on modern newspaper pages.

Table 10: Layout detection results on the out of distribution test set. This table lists F1 Score values for all individual classes. N/A values for Dell et al. (2024) are due to annotation differences. As the OoD data does not contain the `inverted_text` class, it cannot be evaluated in this table. Since YOLOv8 detects bounding boxes, we do not require seperator detection. (Main results for layout in Table 4)

| | F1 Score (OoD only) | | |
|---|---|---|---|
| class | ours | YOLOv8 | Dell et al. |
| background | $0.89 \pm 0.022$ | $0.90 \pm 0.001$ | 0.82 |
| caption | $0.39 \pm 0.132$ | $0.62 \pm 0.092$ | N/A |
| table | $0.67 \pm 0.036$ | $0.24 \pm 0.098$ | 0.16 |
| paragraph | $0.90 \pm 0.025$ | $0.91 \pm 0.012$ | 0.84 |
| heading | $0.50 \pm 0.052$ | $0.63 \pm 0.020$ | 0.43 |
| header | $0.40 \pm 0.085$ | $0.62 \pm 0.033$ | 0.07 |
| separator vertical | $0.58 \pm 0.028$ | N/A | N/A |
| separator horizontal | $0.57 \pm 0.043$ | N/A | N/A |
| image | $0.38 \pm 0.121$ | $0.00 \pm 0.000$ | N/A |
| inverted text | N/A | N/A | N/A |

Table 11: Layout detection results. This table lists F1 Score values for all individual classes form a YOLOv8 detection model trained on our dataset. Since YOLOv8 detects bounding boxes, we do not require seperator detection.

| class | F1 Score (iD + OoD) | F1 Score (iD only) | F1 Score (OoD only) |
|---|---|---|---|
| background | $0.90 \pm 0.003$ | $0.90 \pm 0.003$ | $0.90 \pm 0.001$ |
| caption | $0.79 \pm 0.135$ | $0.80 \pm 0.144$ | $0.62 \pm 0.092$ |
| table | $0.75 \pm 0.020$ | $0.81 \pm 0.027$ | $0.24 \pm 0.098$ |
| paragraph | $0.94 \pm 0.003$ | $0.94 \pm 0.002$ | $0.91 \pm 0.012$ |
| heading | $0.72 \pm 0.013$ | $0.73 \pm 0.015$ | $0.63 \pm 0.020$ |
| header | $0.78 \pm 0.023$ | $0.80 \pm 0.031$ | $0.62 \pm 0.033$ |
| image | $0.03 \pm 0.003$ | $0.03 \pm 0.004$ | $0.00 \pm 0.000$ |
| inverted text | $0.00 \pm 0.000$ | $0.00 \pm 0.000$ | N/A |

### A.7.2 LAYOUT LIMITATIONS

The full layout results (Table 7) show the continued challenge of differentiating between different text regions, especially in OoD data. This is due to strongly varying layout in different newspapers and the lack of sufficiently many and diverse examples in rare classes within our dataset. For the `caption` `header` classes, we observe a drastic drop in performance from iD to OoD data. One common error is, to confuse `header` and `headline` classes, as shown in Figure 11. For `caption` and `header` the limited receptive field of our layout model (Oliveira et al. (2018)) is an issue, as they are mostly detectable by considering the overall position on the newspaper page. We aim to improve this by either employing a CNN-based model with a much higher receptive field or using a detection model. Table 10 contains OoD only results for our layout model, the finetuned YOLOv8, and the Dell et al. layout model. The finetuned YOLOv8 shows better generalization performance than our model on `header`, `heading`, and `caption` classes. However, the results for `table` are much worse than our model. Overall, we aim to further expand our dataset in the future, specifically increasing rare class examples.

Layout errors commonly cause text lines to be split into different text region classes, as shown in Figure 10. Therefore, the pipeline will generate separate text line polygons for one ground truth text line. Another issue are tables, or part thereof, that are being mistreated as text regions, as shown in Figure 8 and Figure 9. In the ground truth data tables do not have line polygons or text annotations. When the pipeline produces text lines in ground truth tables, those lines cannot be matched.

This impacts the matching of text lines during evaluation and has a big effect on the overall pipeline performance. We aim to change the architecture of our pipeline in a way, that prevents text lines from being split by layout borders. Ideally, the split text lines in the left example of Figure 10 would be considered to be about 80% caption and 20% paragraph instead of being split at that region border. However, some layout information has to be considered when detecting baselines as they cannot be drawn from one column to another, ignoring column borders or even separators as shown in Figure 6. This might be achievable by splitting the layout segmentation task up in a basic text region detection task and a segmentation task, that differentiates between different text regions. Baseline detection has to be independent of the latter and dependent on the former.

### A.7.3 BASELINE LIMITATIONS

Figure 12 shows examples of baseline errors due to changing font size within text lines. Such examples occur almost exclusively in advertisements, therefore being very rare in our data. Our baseline detection model fails in correctly connecting baselines in such cases. We aim to add additional data containing drop capitals and font changes.

### A.7.4 ARTICLE SEPARATION

Article Separation is of great importance to enable further processing of transcribed pages. The main challenge lies in articles that span over page breaks and require the consideration of neighbouring pages. It is possible for multiple articles to span over the same page break. It has to be considered, that some newspaper pages are digitized in a way, that includes an entire double page in one image. We aim to create ground truth data for page spanning article separation and explore approaches to separate articles automatically.

### A.7.5 ENABLING HISTORICAL RESEARCH

Ideally, our work will enable the processing of millions of pages of historical data, making vast resources easily available to future researchers who can then build upon the transcribed source material, for example, with machine translation and Natural Language Processing (NLP) pipelines. Countless research questions concerning economic, societal, political and scientific development can be addressed with such data. For a more detailed description of the relevance of such data for

historical research, see Supplementary Section A.2 . We hope this dataset will help to improve our understanding of the past.

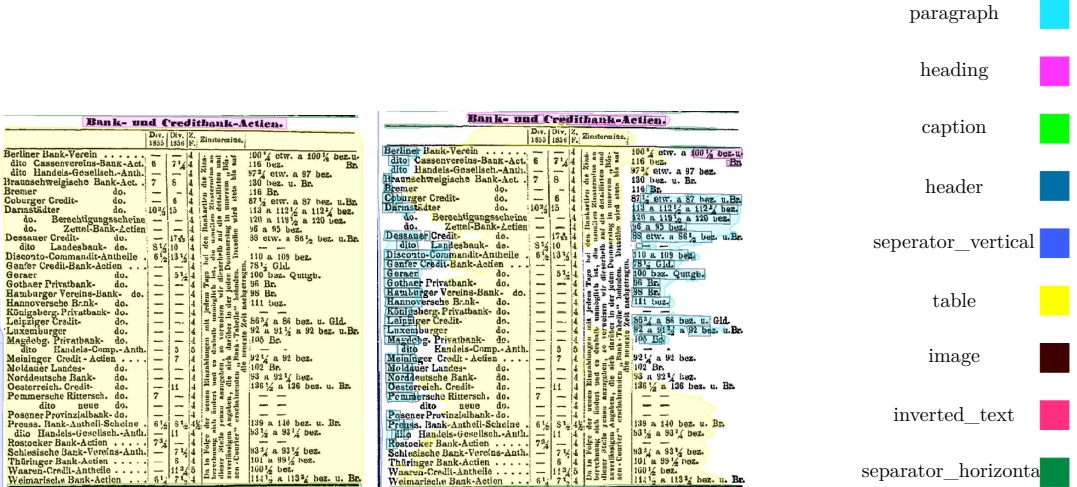

Figure 8: Target labels on the left and segmentation prediction on the right. Challenging table example from the iD test set.

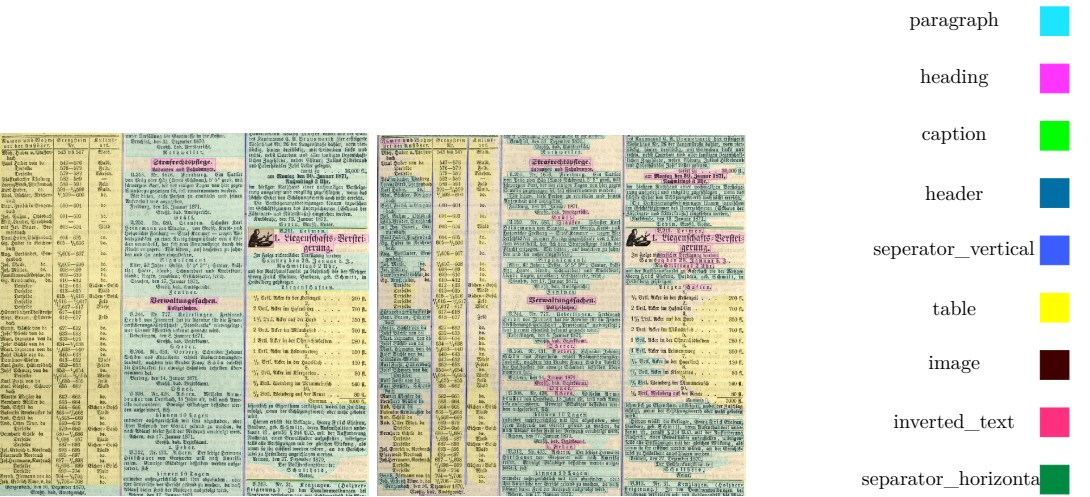

Figure 9: Target labels on the left and segmentation prediction on the right. Challenging table example from the OoD test set.

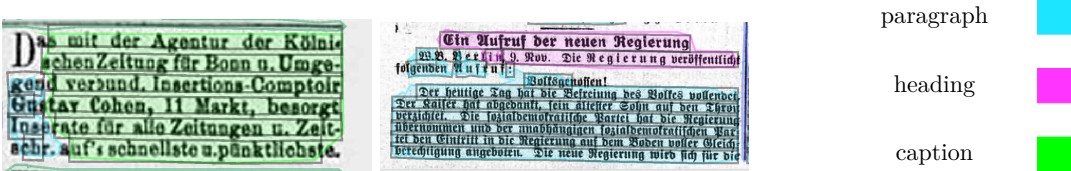

Figure 10: Examples for split text lines due to layout errors.

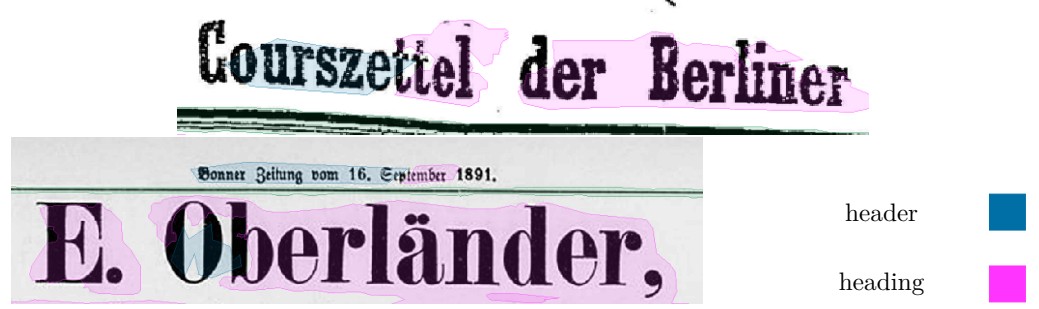

Figure 11: Layout Prediction Examples for the challenge of differentiating headlines and header, which both have a higher font size than usual text.

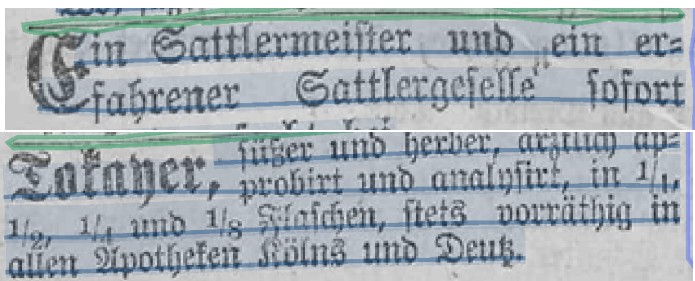

Figure 12: Examples for baseline errors caused by changing font sizes within text lines. On top is a drop capital that is the first character of the first line. The lower example is an entire word written in a bigger font. The sentence continues on the upper text line to the right of it.

