# OpenReview forum: "Chronicling Germany: An Annotated Historical Newspaper Dataset"
_DMLR — Accepted by DMLR_

### Review · Reviewer_pQ3R · 2025-04-03

**Recommendation:** 4
**Confidence:** 2

**Summary Of Contributions:**

This paper presents "Chronicling Germany," a dataset consisting of 693 annotated historical newspaper pages spanning from 1852 to 1924. This dataset represents a unique and highly valuable resource, beneficial not only to computer scientists / machine learning but also to historians and other interdisciplinary researchers. Besides the contribution of the dataset, the paper also presents a processing pipeline.

The dataset, from my personal point of view, is a large dataset and a lot of work, considering the amount of texts on each page of the newspaper and efforts from domain experts. This paper serves as an exemplary model of interdisciplinary collaboration between computer science and the liberal arts, laying an important foundation for future research on collecting, cleaning, and preparing similar datasets, particularly for low-resource European languages. I support the acceptance of this paper.

**Strengths:**

See above ^^

**Audience:**

Yes

**Broader Impact Concerns:**

No immediate impact concerns.

**Claims And Evidence:**

Yes as far as I can tell.

**Datasets And Benchmarks:**

Yes it includes sufficient details, and the data is also made available.

**Extended Submissions:**

It's is not an extended version of a previously published work as far as I can tell.

**Limitations:**

See above ^^

**Requested Changes:**

* The chosen timeframe (1852–1924) seems somewhat limited for a dataset titled "Chronicling Germany." Do the authors plan to extend the dataset's timeframe in the future, possibly covering a broader historical period (e.g., 1925–1945)? I suggest revising the paper title (Maybe "Chronicling Germany -- 1852 to 1924") to better reflect its current temporal scope if no immediate expansion is intended. Personally, I am particularly interested in newspapers covering the period from 1924 to 1945 (especially during the World War era).
* The authors mention the Neue Züricher Zeitung (NZZ) dataset. Out of curiosity, and to clarify (maybe I missed something): How well does the proposed pipeline, trained specifically on German newspapers, perform when applied to the NZZ dataset? Demonstrating strong generalization performance on newspapers from other German-speaking countries could significantly enhance the paper's contribution and serve as a valuable artifact. I think it would be nicer change and will make this paper stronger.

**Strengths And Weaknesses:**

- The dataset itself is interesting, challenging to collect and clean, and I imagine this is a lot of efforts. The dataset is also made publicly available. It is highly relevant to DMLR.
- The paper is well written and clearly explained the challenges and techniques they used. I like the flowcharts (in Figure 2) and in general how authors present.
- This work is broadly relevant, not only to computer scientists but also to historians and potentially the general public, underscoring its interdisciplinary value.

---

### Review · Reviewer_zrsp · 2025-04-09

**Recommendation:** 3
**Confidence:** 3

**Summary Of Contributions:**

The author proposed a dataset designed to enable the training of layout and OCR models for historic
German-language newspapers in which 693 manually annotated high-resolution pages are publicly available. The authors also developed a
recognition pipeline for the layout detection, text-line recognition, and OCR-tasks.

**Strengths:**

The submission is of good readability, easy to follow, and of good clarity. Prior works and related datasets are discussed in details. This dataset is of broad interest as it enables research on layout segmentation, ML-based OCR and NLP.

**Audience:**

Yes

**Claims And Evidence:**

Yes, claims made in the submission are supported by accurate and convincing experimental result.

**Datasets And Benchmarks:**

The dataset is described in details and publicly available. Technical details are discussed for experiment reproducibility.

**Extended Submissions:**

N/A

**Limitations:**

The limitations are also discussed in the paper, e.g., networks trained exclusively using the proposed dataset are not likely to outperform more specialized networks on modern newspaper pages due to Fraktur-letters. Other limitations include recognition of the drop-capitals present in advertisement pages and abbreviations and fractions in the market and stock exchange reports of the newspapers.

**Requested Changes:**

It would be better to explain, in Section 4.1 when comparing proposed pipeline with the one that is developed by Dell et al. (2024), why this benchmark model was not fine tuned with this dataset.

**Strengths And Weaknesses:**

Strengths:

1. Both dataset and code base are publicly available.
2. This is the largest annotated collection of historic German newspaper pages.
3. The details of implementation of experiments are described in the paper.

Weakness:

1. contains some typos, e.g., "...share processing coda....."

---

### Review · Reviewer_6Cnu · 2025-04-25

**Recommendation:** 3
**Confidence:** 2

**Summary Of Contributions:**

The paper introduces a new dataset consisting of 693 annotated historical newspaper pages from Germany spanning the period between 1852 and 1924. This dataset specifically targets enhancing Optical Character Recognition (OCR) and layout detection models for historical German newspapers, addressing challenges posed by the dense layouts and distinct typefaces typical of historical documents. The paper also releases datasets and code under open repositories to stimulate future research.

**Strengths:**

- The proposed dataset addresses a pressing need for high-quality German Fraktur resources, beneficial to both computer vision researchers and digital historians.

- The authors establish a clear baseline performance using well-known model architectures, providing a reliable benchmark for future improvements.

- Open access to both the dataset and the accompanying code greatly facilitates replication and future experimentation.

**Audience:**

Yes

**Broader Impact Concerns:**

No concern.

**Claims And Evidence:**

Yes

**Datasets And Benchmarks:**

Yes

**Extended Submissions:**

N/A

**Limitations:**

See above.

**Requested Changes:**

- Although the dataset undeniably holds value for historians and social scientists, given that the submission is to a machine learning venue, it is essential to clearly articulate the dataset's relevance and benefits from a machine learning perspective. The paper briefly mentions potential use as a low-resource task for computer vision; however, creating low-resource CV tasks is relatively straightforward and has been explored in existing literature. I suggest providing a more focused and detailed rationale for the dataset. Specifically, does it support a novel task or address a previously unexplored challenge in CV? How does this dataset differ from existing benchmarks, and what unique capabilities or evaluation opportunities does it offer that other datasets do not?

- The paper notes challenges in accurately predicting rare classes, such as images and inverted text. Further exploration into tailored data augmentation techniques or specialized loss functions designed explicitly for handling class imbalance could significantly enhance model robustness.

- The manuscript does not mention the dataset's licensing or copyright status. Clarifying this information is critical for ensuring its usability and distribution in downstream applications.

**Strengths And Weaknesses:**

**Strengths:**

- The proposed dataset addresses a pressing need for high-quality German Fraktur resources, beneficial to both computer vision researchers and digital historians.

- The authors establish a clear baseline performance using well-known model architectures, providing a reliable benchmark for future improvements.

- Open access to both the dataset and the accompanying code greatly facilitates replication and future experimentation.


**Weaknesses:**

Thank you for submitting this paper to DMLR. I appreciate the authors' significant efforts in dataset collection and have no major concerns regarding the quality or data collection methodology. My primary critique relates to the paper's motivation and positioning within the machine learning community. Detailed comments follow:

- Although the dataset undeniably holds value for historians and social scientists, given that the submission is to a machine learning venue, it is essential to clearly articulate the dataset's relevance and benefits from a machine learning perspective. The paper briefly mentions potential use as a low-resource task for computer vision; however, creating low-resource CV tasks is relatively straightforward and has been explored in existing literature. I suggest providing a more focused and detailed rationale for the dataset. Specifically, does it support a novel task or address a previously unexplored challenge in CV? How does this dataset differ from existing benchmarks, and what unique capabilities or evaluation opportunities does it offer that other datasets do not?

- The paper notes challenges in accurately predicting rare classes, such as images and inverted text. Further exploration into tailored data augmentation techniques or specialized loss functions designed explicitly for handling class imbalance could significantly enhance model robustness.

- The manuscript does not mention the dataset's licensing or copyright status. Clarifying this information is critical for ensuring its usability and distribution in downstream applications.